# Variation of Structural and Dynamical Flexibility of Myelin Basic Protein in Response to Guanidinium Chloride

**DOI:** 10.3390/ijms23136969

**Published:** 2022-06-23

**Authors:** Luman Haris, Ralf Biehl, Martin Dulle, Aurel Radulescu, Olaf Holderer, Ingo Hoffmann, Andreas M. Stadler

**Affiliations:** 1Jülich Centre for Neutron Science (JCNS-1) and Institute of Biological Information Processing (IBI-8), Forschungszentrum Jülich GmbH, 52425 Jülich, Germany; l.haris@fz-juelich.de (L.H.); ra.biehl@fz-juelich.de (R.B.); m.dulle@fz-juelich.de (M.D.); 2Institute of Physical Chemistry, RWTH Aachen University, Landoltweg 2, 52056 Aachen, Germany; 3Jülich Centre for Neutron Science (JCNS) at Heinz Maier-Leibnitz Zentrum (MLZ), Forschungzentrum Jülich GmbH, 85747 Garching, Germany; a.radulescu@fz-juelich.de (A.R.); o.holderer@fz-juelich.de (O.H.); 4Institut Laue-Langevin, 71 Avenue des Martyrs, CS 20156, CEDEX 9, 38042 Grenoble, France; hoffmann@ill.fr

**Keywords:** myelin basic protein, guanidinium chloride, small-angle scattering, neutron spin–echo spectroscopy, internal protein dynamics, polymer theory, normal mode analysis

## Abstract

Myelin basic protein (MBP) is intrinsically disordered in solution and is considered as a conformationally flexible biomacromolecule. Here, we present a study on perturbation of MBP structure and dynamics by the denaturant guanidinium chloride (GndCl) using small-angle scattering and neutron spin–echo spectroscopy (NSE). A concentration of 0.2 M GndCl causes charge screening in MBP resulting in a compact, but still disordered protein conformation, while GndCl concentrations above 1 M lead to structural expansion and swelling of MBP. NSE data of MBP were analyzed using the Zimm model with internal friction (ZIF) and normal mode (NM) analysis. A significant contribution of internal friction was found in compact states of MBP that approaches a non-vanishing internal friction relaxation time of approximately 40 ns at high GndCl concentrations. NM analysis demonstrates that the relaxation rates of internal modes of MBP remain unaffected by GndCl, while structural expansion due to GndCl results in increased amplitudes of internal motions. Within the model of the Brownian oscillator our observations can be rationalized by a loss of friction within the protein due to structural expansion. Our study highlights the intimate coupling of structural and dynamical plasticity of MBP, and its fundamental difference to the behavior of ideal polymers in solution.

## 1. Introduction

Intrinsically disordered proteins (IDPs) are exceptionally rich in their biological functions despite being structurally disordered to a large extent [1]. Their inherent flexibility constitutes a central element for their various biological tasks in which they are involved [2]. IDPs are exceptionally rich in polar and charged residues in comparison to globular proteins [3,4], which results in a plethora of structural states that are separated by low energy barriers [5]. These characteristics allow IDPs to dynamically sample a huge conformational ensemble by means of large-scale correlated motions of the protein chain [6,7] that are connected to backbone torsional rotations occurring on short length ranges [8,9]. Concurrently, IDPs are highly susceptible in their structural and dynamical response to external perturbations, e.g., variation of ionic strength [10] or solution osmolarity [11], and this constitutes an inherent mechanism of IDPs to respond to local variations of the intracellular medium [12]. Currently, there is still a large gap in knowledge on the response of IDPs to variation of solvent conditions and on the intricate balance between IDP structure and dynamics.

One prominent example of IDPs is myelin basic protein (MBP) that works in tandem with cholesterol to maintain the integrity of the myelin sheath [13,14]. The functionality of MBP is closely related to its high degree of structural flexibility that depends sensitively on specific solvent conditions such as pH value, dielectric strength of the solvent medium or the interaction with membranes [15,16,17,18,19,20,21,22,23,24,25]. MBP bears a net positive charge at physiological conditions. The positive net charge of MBP is responsible not only for its extended conformation in solution, but also its biological function as MBP interacts electrostatically with negatively charged myelin lipids and acts as a “molecular glue” for the myelin membrane sheath [14,20].

In this study, we report on the influence of the strong chaotropic denaturant guanidinium chloride (GndCl) on structure and dynamics of MBP in solution. Small-angle scattering by X-rays and neutrons (SAXS/SANS) as well as neutron spin–echo spectroscopy (NSE) have been used to provide information on structure and dynamics of MBP, respectively, in response to GndCl concentration. The combination of those techniques has been applied previously for the investigation of unfolded and partially folded proteins in solution [21,22,26,27] revealing the intricate connection between structural constraints on protein fluctuations.

## 2. Results and Discussion

Structural properties of MBP in response to GndCl concentration changes have been investigated as underlying basis for further NSE studies regarding protein dynamics. For neutron scattering experiments —and, in particular, for NSE studies—deuterated solvents are required to reduce the incoherent scattering contribution of the background and to maintain the polarization of the neutron beam in the NSE instrument. For consistency, all experiments have been performed at a sample temperature of 295 K and in D_2_O buffer (99.9% D atom, 50 mM Na_2_HPO_4_/NaH_2_PO_4_, pD 7.0) containing additional different concentrations of the denaturant GndCl. In the following section, we first present the results of circular dichroism (CD) measurements, followed by a structural characterization of MBP based on SAXS/SANS as well as dynamic light scattering (DLS) experiments.

### 2.1. Structural Investigation Using CD and SAXS/SANS Experiments

Secondary structure content of MBP was investigated using CD. The CD data as presented in Figure 1A demonstrate that MBP is partially unfolded under all investigated GndCl concentrations from 0 M to 6 M GndCl. In the absence of denaturant, MBP appears to be largely disordered indicated by a global minimum slightly above 200 nm (Figure 1A). The fact that the position of the minimum is slightly above 200 nm indicates the presence of folded secondary structure content such as α-helix or β-sheet. Interestingly, similar behavior can also be observed with increasing GndCl concentration up to 6 M GndCl (Figure 1A). Quantitative estimation of the secondary structure content has been done using Dichroweb [28,29]. We found that the disordered fraction amounts to ~55% for all investigated GndCl concentrations. These observations agree with recent results based on synchrotron radiation CD and secondary structure estimation that yielded a similar fraction of unfolded secondary structure of MBP in D_2_O-based phosphate buffer in the absence of GndCl at slightly acid pH [23]. Furthermore, no change of MBP secondary structure content or a cooperative unfolding transition has been observed in that study up to 360 K [23]. Hence, we can conclude that MBP remains a partially folded protein irrespectively of temperature or GndCl concentration.

Upon further observation, a peculiar peak at around 218-222 nm can be seen. It becomes more evident with increasing GndCl concentration (Figure 1B). Previous studies by Polverini et al. [30] and Vassall et al. [16], showed that MBP contains a proline rich region that apparently plays an important role in protein-ligand binding. This region forms a structure commonly known as polyproline type II (PPII) helix in MBP under physiological conditions [16,30]. The PPII helix is prevalent in unfolded or disordered proteins [31]. It is inherently more flexible than α-helix or β-sheet and contributes to the flexibility and expansion of the whole protein structure [31]. Interestingly, the PPII structure is found to be stabilized by low temperature [30] or high amount of GndCl concentration [32,33]. The latter case contrasts with the standard observation for globular folded proteins that show loss of secondary structure content and formation of random coil at high GndCl concentration [34]. To follow the signature reminiscent of the PPII helix in MBP, we plotted the CD signal at 222 nm as function of GndCl concentration (see Figure 1B). We observe the emergence of the PPII helix signal above 2 M GndCl and, for higher GndCl concentrations, it is fully established. We interpret this observation by CD as evidence for the progressive expansion of MBP for GndCl concentrations larger than 2 M GndCl albeit the disordered fraction of ~55% remains unchanged. This observation strongly supports our further structural investigations as we will see in the next paragraph.

The structure of MBP was investigated with SAXS and SANS: X-ray scattering experiments were performed using a lab-based Ganesha SAXS instrument (Xenocs, Grenoble, France) located at JCNS-1/IBI-8, FZJ, Jülich, Germany and SANS was measured on the small-angle neutron diffractometer KWS-2 located at MLZ, Garching, Germany [35,36,37]. Selected SAXS form factors $I\left(q\right)/c$ at a protein concentration of c = 0.5% *w*/*v* are shown in Figure 2A (see Appendix A for SAXS data of all GndCl concentrations and Appendix A for SANS data). In general, SAXS data of MBP are comparatively noisy for GndCl concentrations of 4 M and 6 M. This is due to the loss of contrast and strong absorption of X-rays at high GndCl concentrations. Hence, SANS experiments have been performed of MBP in 1, 4 and 6 M GndCl to confirm the observations by SAXS and to validate the results. SAXS and SANS form factors of MBP are well described by generalized Gauss functions (see Figure 2A, Appendix A) yielding radii of gyration *R*_G_ and power law coefficients *d* describing the scattering behavior with ~*q*^−*d*^ for high *q*-values. The inverse of the power law scaling coefficient gives rise to a scaling exponent ν=1/d that describes chain statistics of the biomacromolecule. Obtained values of *R*_G_ and ν from fits using generalized Gauss functions are reported in Figure 2C,E. Guinier radii were also calculated from the experimental SAXS/SANS form factors using the Guinier approximation, which yields *R*_Guinier_ values that are independent of the analytical model being used. The *R*_Guinier_ values obtained from the Guinier analysis are reported in Figure 2C as well. Guinier fits of SAXS and SANS data are shown in Appendix A. Within the errors, the *R*_Guinier_ values from the Guinier analysis agree with the corresponding ones from the generalized Gauss fits, which demonstrates the overall validity of our SAXS/SANS data analysis approach.

GndCl is well known for its preferential affinity for hydrophobic residues and folded secondary structure such as α-helices [38]. Typically, a transition from a stabilizing effect at low GndCl concentration to denaturation at higher GndCl concentration has been observed for folded globular proteins [39,40]. On the other hand, concerning IDPs a transition from a highly expanded structural conformation in 0 M GndCl to a structurally more collapsed one due to charge screening effects within the IDPs by GndCl has been observed previously by single-molecule spectroscopy [6,10]. This effect becomes more pronounced for IDPs carrying a larger absolute value of the charge. In general, further increase of GndCl concentration results in progressive swelling and structural expansion of IDPs as confirmed by a variety of different experimental techniques [41].

Concerning MBP, we find a slight reduction of the radius of gyration *R*_G_ of MBP at 0.2 M GndCl as compared to its *R*_G_-value at 0 M GndCl (see Figure 2C), while further addition of GndCl results in structural expansion of MBP as evidenced by the increase of the *R*_G_-values. The fully expanded state of MBP is reached at 4 M and 6 M GndCl. This swelling behavior is also visible in the scaling exponents that increase from values of ν=0.48 at 0 M GndCl and ν=0.49 at 0.2 M GndCl to larger values of ν ~ 0.53 for c_GndCl_ at 1 M and above. The behavior of both *R*_G_ and ν reveal a more compact structure of MBP at low GndCl concentrations and progressively more expanded configurations at high GndCl concentrations. The configurations of GndCl in high GndCl concentrations are, however, still significantly more compact than denatured globular proteins, which are characterized by a value of ν=0.598±0.028 [42]. The latter are close to ideal excluded volume polymers that show ν=0.588 [43]. From a polymer-based perspective, the ν-values reveal the transition from a Gaussian-like and disordered protein chain configuration of MBP at 0 M and 0.2 M GndCl to an expanded and swollen one where excluded-volume effects are prevailing. This structural expansion of MBP is also discernible in the inflection point at approximately 2 M GndCl of the CD data (Figure 1B), which we attribute to the structural expansion of MBP at high GndCl concentrations.

In an alternative approach, the SAXS form factors of MBP have been analyzed using the worm-like chain model [44], which has been used classically for the description of semiflexible polymers with excluded volume interactions. Selected Kratky plots are shown in Figure 2B with fits using the worm-like chain model [44]. The Kratky plots show a plateau of MBP in 0 M and 0.2 M GndCl, which is a characteristic sign for IDPs with Gaussian-like disordered chains [45]. Deviations from that plateau at larger *q*-values are related to the persistence length of the protein chain. The fully denatured and expanded state at high denaturant concentrations—such as for the shown SAXS data at 6 M GndCl—leads to strong deviations of the plateau and the flexible nature of MBP under those conditions becomes clearly visible. The structural features of MBP are essentially captured by the worm-like chain model: The worm-like chain model describes MBP as a series of cylinder-shaped segments. Each of which has a radius *R* and a length defined by the Kuhn length. Half of the Kuhn length thus gives rise to the persistence length *l*_p_, which is often used in the quantification of chain flexibility. The total contour length of the chain was set to the number of residues * distance between C_α_ atoms (169 * 0.38 nm = 64.2 nm). Only the persistence length *l*_p_ and the cylinder radius *R* have been fitted. Values of *l*_p_ are reported in Figure 2F (see Appendix A for the other parameters). The persistence length is slightly reduced at 0.2 M GndCl as compared to its initial value at 0 M GndCl, whereas it increases again at further GndCl addition. The length scale of the stiff segment in reciprocal space is given by *q** = 1.91/*l_p_* [46], which is indicated in Figure 2B by asterisks.

Hydrodynamic radii *R*_H_ were measured by using dynamic light scattering (DLS) at a protein concentration of 0.5 % *w*/*v* (see Appendix A for experimental parameters). The ratio of *R*_H_/*R*_G_ has been used as sensitive indicator for the compactness of a protein, where $\sqrt{5/3}=1.29$ and 0.67 are the known extreme ratios for a compact globular protein and a random-coil-like protein, respectively [47,48]. The *R*_H_/*R*_G_ values of MBP for the GndCl concentrations considered in this work are well within these two extremes see Figure 2D, thus implying a partially disordered conformation at 0 M GndCl, a slightly more collapsed but still disordered state of MBP at 0.2 M GndCl due to charge screening by GndCl, and significantly more extended MBP conformations above 1 M GndCl due to GndCl induced swelling and structural expansion.

Structure factors *S*(*q*) of concentrated 5% *w*/*v* MBP solutions have been obtained from SAXS and SANS experiments and are shown in the inset of Figure 2A. Structure factors of MBP in 0 M and 0.2 M GndCl were fitted effectively using a model for charged hard spheres to consider attractive patchy protein-protein interactions that are visible by the upturn in the *S*(*q*) for q→0. At *c*_GndCl_ = 1 M the structure factor is approximately *q*-independent, which demonstrates the effective absence of protein-protein interactions. In the field of polymer physics this case would correspond to the so called θ-state, an ideal situation where inter- and intra-chain interactions are effectively absent. For high GndCl concentrations of 4 M and 6 M the electrostatic interactions are fully screened, and the structure factors were fitted with a model for non-charged hard-sphere particles. Protein–protein interactions are repulsive under those conditions and dominated by excluded volume interactions. Fitted parameters are compiled in Appendix A. The experimentally determined structure factors *S*(*q*) have been used for the analysis of NSE data as described further below.

An inverse Monte Carlo approach was applied to gain further insights in the conformational ensemble of MBP and on the effect of GndCl on protein structural plasticity. Ensemble optimization modelling (EOM) [49] as available within the ATSAS software suite was used to generate first a large set of coarse-grained structural models of MBP; in a second step a genetic algorithm was applied to select conformational ensembles based on direct comparison with experimental SAXS data. Final fits to the SAXS data using the EOM modelling approach and obtained radii of gyration *R*_G,EOM_ distributions from EOM modelling are shown in Figure 3A,B. Due to the larger errors of the SAXS data at 6 M GndCl, the fitted generalized Gauss function was used as restraint for EOM modelling, see Figure 3A. Similar distributions were obtained when experimental SAXS data of MBP in 6 M GndCl was used (data not shown). The *R*_G,EOM_ distributions inform on the structural plasticity of MBP: At 0 M and 0.2 M GndCl, progressively narrower *R*_G,EOM_ distributions are obtained, while increased GndCl concentrations result in broader distributions due to swelling of MBP and structural expansion. These observations by EOM analysis support the results discussed above. Most frequently occurring coarse-grained models during EOM modelling are shown in Figure 3C. These structural models have been used for the calculation of collective low-frequency normal modes to describe internal protein dynamics of MBP as discussed in the next section further below. Information regarding the radii of gyration *R*_G,EOM_ and maximal dimensions *D*_max_ of coarse-grained EOM models and their corresponding population fractions are compiled in Appendix A.

### 2.2. Investigation of Protein Dynamics Using NSE

NSE enables the observation of collective protein dynamics from the sub-nanosecond up to several hundred nanoseconds and on a nanometer length scale [50,51]. Dynamics of MBP has been measured on the NSE spectrometers IN15 (ILL, Grenoble, France) [52] in 0 M and 0.2 M GndCl and J-NSE (MLZ, Garching, Germany) [53] in 1 M, 4 M, and 6 M GndCl at a protein concentration of 5% *w*/*v*. Representative intermediate scattering functions $I\left(q,t\right)/I\left(q,0\right)$ of MBP in 0.2 M and 6 M GndCl are shown in Figure 4A,B including fits using the theoretical Zimm and Zimm with internal friction (ZIF) models that are derived from polymer theory [47,54,55]. Additionally, an alternative interpretation of the NSE spectra that is based on normal mode (NM) analysis using the structural ensemble generated by EOM modelling is shown in Figure 4C,D [21,22]. NSE data analysis and results based on Zimm/ZIF modelling and NM analysis are discussed further below in detail. The complete NSE data sets of all investigated GndCl concentrations including theoretical Zimm/ZIF modelling as well as NM analysis are shown in Appendix A, respectively.

The NSE relaxation spectra are generally well described by stretched exponential functions (see Appendix A), which yield *q*-dependent effective diffusion coefficients *D*_eff_(*q*) and stretching coefficients β. Averaged over all *q*-values the stretching coefficients are 〈β〉≈0.91−0.93 for *c*_GndCl_ ≤ 0.2 M, and 〈β〉≈0.86 for *c*_GndCl_ ≥ 1 M (see Appendix A). The values of the stretching coefficients 〈β〉 are in the range that is expected for Gaussian polymers in solution following ideal Zimm dynamics ($\beta_{Zimm}=0.85$) [53,54]. As can directly be seen by the stretching coefficients, dynamics of MBP is close to Zimm-like behavior in 1 M, 4 M and 6 M GndCl, where MBP is in its expanded and swollen state. Motions of MBP in 0 M and 0.2 M GndCl are closer to simple exponential behavior. The underlying reason for that is the larger contribution of global protein diffusion to the observed dynamics as seen by NSE for the more compact MBP structures in the low GndCl concentration range [22,56]. Similar reduction of internal motions has previously been observed in dynamically rigid polymers due to Zimm-mode reduction [57]. Whilst different dynamic behavior of MBP is already clearly discernible in the model-free data analysis approach, we focus on a quantitative interpretation of the NSE spectra in the next sections.

Since NSE measures the relaxation process covering a large range of internal modes the Zimm model was considered because of the similarity of IDPs with polymers. The Zimm model considers the IDP as coarse-grained beads that are connected by entropic springs considering hydrodynamic interactions. The Zimm model yields a mode dependent relaxation with characteristic time $\tau_{p}$ given by
(1)$$\tau_{p}=\frac{\eta R_{e}^{3}}{\sqrt{3\pi }k_{B}T}p^{-3\nu }$$
where η is the solvent viscosity, *R**_e_* the end-to-end distance of the chain, $k_{B}$ the Boltzmann constant, *T* temperature, and *p* the Zimm mode number [44]. The end-to-end distance is given by $R_{e}=\sqrt{\left(2\nu +1\right)∙\left(2\nu +2\right)}\;R_{G}\;$ [58,59], where $R_{G}$ and ν have been obtained directly from the generalized Gauss fits of the SAXS data (see Figure 2C,E and Appendix A). The Zimm analysis used 20 beads which correspond to 8 amino acids per bead in average. Each of the beads is separated by a bond length $l=R_{e}/N^{\nu }$. Finally, the intermediate scattering function of the Zimm model is given by
(2)$$I\left(q,t\right)=\frac{\exp\left(-q^{2}D_{t}t\right)}{N}\overset{N}{\underset{n,m}{\sum }}exp\left[\frac{-q^{2}B\left(n,m,t\right)}{6}\right]$$
(3)$$B\left(n,m,t\right)={\left(n-m\right)}^{2\nu }l^{2}+\frac{4R_{e}^{2}}{\pi^{2}}\overset{p_{max}}{\underset{p=1}{\sum }}\frac{1}{p^{2\nu +1}}\cos\left(\frac{\pi pn}{N}\right)\cos\left(\frac{\pi pm}{N}\right)\left[1-\exp\left(-\frac{t}{\tau_{p}}\right)\right]$$
where the maximum mode number *p**_max_* is the same as the number of beads *N* and $\tau_{p}$ is the characteristic Zimm time of mode *p* [47]. The translational diffusion *D**_t_* is corrected for hydrodynamic and interparticle interactions by [22,50,51]
(4)$$D_{t}\left(q\right)=D_{0,t}H/S\left(q\right)$$

The single protein translational diffusion $D_{0,t}$ was determined by DLS measurements of 0.5% *w*/*v* solutions (see Appendix A). The structure factor *S*(*q*) was obtained experimentally using SAXS/SANS (see inset Figure 1A), and *H* was assumed to be a *q* independent fit parameter. Results regarding hydrodynamic interactions are shown in Figure 5.

The Zimm model does not consider interactions on length scales that are smaller than the bond length. These contributions include, for instance, interactions between side chains, H-bonding, backbone dihedral angle barriers and other types of local steric hindrances and interactions. These effects effectively lead to the suppression of higher Zimm modes. They are in practice considered by adding a resistive damping to the bead connecting springs of the Zimm model, yielding the so-called Zimm model with internal friction (ZIF) [54]. This internal friction leads to a modification of the mode dependent Zimm time by $\tau_{p}^{ZIF}=\tau_{p}+\tau_{i}$, where $\tau_{i}$ is the additional relaxation time due to internal friction. Due to the coarse-grained nature of the Zimm/ZIF models, our choice for the bead number *N* = 20 proved to be accurate and the results remained unaffected by increasing the bead number in the ZIF fits due to the underlying nature of scale invariance of the polymer models (compare Appendix A for an overview of the χ^2^-values informing on the goodness of fit of the different models that were used in this study).

NSE spectroscopy is a unique technique that allows to probe protein short-time diffusion where the root mean square displacement (RMSD) due to protein diffusion is smaller than the protein size and the cage of next neighbor proteins is not changed. The corresponding hydrodynamic function $H_{short}$ is informative on hydrodynamic interactions between individual MBP proteins. We find (see Figure 5) that $H_{short}$ of the ZIF modelling approach is larger for c_GndCl_ ≥ 1 M GndCl than what would be expected from viscometry experiments that inform about solution behavior on macroscopic length and time scales, the long-time diffusion regime according to $H_{long}=\eta_{0}/\eta$ where $\eta_{0}$ and η are the measured viscosities of the buffer and protein solutions, respectively [60]. For $H_{short}$ of the ZIF model both the change of protein configuration to the swollen and expanded state as well as the transition from attractive MBP-MBP interactions at 0 and 0.2 M GndCl to excluded-volume repulsion for c_GndCl_ > 1 M GndCl seem to be the important factors leading to the deviation from $H_{long}$. Similar observations have been made by Bucciarelli et al. [61] who observed significant slowing down of protein short-time diffusion due to attractive patchy protein-protein interactions as compared to the case where protein interactions are from the non-charged hard-sphere type.

The comparison of both Zimm and ZIF models with respect to the experimental NSE data are shown in Figure 4 for representative GndCl concentrations of 0.2 M and 6 M and in Appendix A for all investigated GndCl concentrations. The Zimm model fails to properly describe the NSE spectra, which is clearly visible from the fits shown in Figure 4. This is especially evident at high *q* where the internal friction dominates the relaxation spectra. The statement holds true for all GndCl concentration considered in this study but is most clearly visible for 0 M and 0.2 M GndCl. Consequently, we obtain a large value of $\chi^{2}=26.5$ for the fits using the Zimm model to the entire NSE data sets of all investigated GndCl concentrations, which demonstrates the poor fit quality. On the other hand, the ZIF model provides with a good description of the entire NSE data set. The goodness of fit parameter of the ZIF model for the entire NSE data set of all GndCl concentrations is reduced to $\chi^{2}=5.2$, which demonstrates quantitatively the significantly improved fit.

Values of $\tau_{i}$ for the different GndCl concentrations are shown in Figure 6A. In the absence of GndCl at neutral pD, MBP is 55% unfolded, and the ZIF model yields $\tau_{i}=69.6\;\pm 5.1$ ns at a sample temperature of 295 K. This is slightly faster than the previously reported value of $\tau_{i}=81.6\pm 3.2$ ns for MBP at 283 K [22]. The weak dependence of $\tau_{i}$ on temperature indicates that internal motions in the intrinsically disordered MBP appear to be regulated by lower activation energies (in the order of ~10 kJ/mol) as compared to unfolded globular proteins: Biehl et al. [26] reported a value of 33 kJ/mol for thermally unfolded ribonuclease A, while on the other hand Xia et al. [62] found indications for temperature independent behavior of internal friction in two model peptides. The most compact conformation of MBP at 0.2 M GndCl yields a slightly larger $\tau_{i}=76.6\;\pm 8.2$ ns that is within the statistical accuracy similar to the result at 0 M GndCl. Internal friction in both 0 M and 0.2 M GndCl exceeds the first Zimm mode (0 M GndCl: $\tau_{p=1}=72.8\;\mathrm{ns}$; 0.2 M GndCl: $\tau_{p=1}=63.6\;\mathrm{ns}$), which is assigned to rotational motion (see Appendix A for a comparison of $\tau_{i}$ and $\tau_{p=1}$ of all samples). Further addition of GndCl appears to reduce internal friction in MBP leading to a limiting minimal $\tau_{i}$ at approximately 40 ns (see Figure 6A) being smaller than the first Zimm mode (1 M GndCl: $\tau_{p=1}=101.9\;\mathrm{ns}$; 4 M GndCl: $\tau_{p=1}=133.0\;\mathrm{ns}$; 6 M GndCl: $\tau_{p=1}=182.4\;\mathrm{ns}$). The large $\tau_{i}$ questions the validity of the ZIF model as in the transition to a rigid chain (approaching large friction with $\tau_{i}\to \infty$) rotational diffusion is not included, which is assigned to the first Zimm mode.

An alternative approach is to describe the observed MBP dynamics by translational and rotational diffusion of a rigid chain complemented by overamped motions along NMs of the respective EOM structures. The dynamics of a single EOM structure is described in this way by [22,50,51]:(5)$$\frac{I\left(q,t\right)}{I\left(q,0\right)}=\left[\left(1-A\left(q\right)\right)+A\left(q\right)e^{-\frac{t}{\tau_{NM}}}\right]∙{\;e}^{-Q^{2}D_{t}\tau }\underset{l=0..\infty }{\sum }S_{l}\left(Q\right)e^{-l\left(l+1\right)D_{r}t}/\underset{l=0..\infty }{\sum }S_{l}\left(Q\right)$$
with
(6)$$S_{l}\left(Q\right)=\sum_{m}{\left|\sum_{i}b_{i}j_{l}\left(Qr_{i}\right)Y_{l,m}\left(\Omega_{i}\right)\right|}^{2}.$$

*S*_l_(*Q*) are the terms of a multipole expansion with scattering length *b**_i_* of the atom *i* at position *r**_i_* and orientation $\Omega_{i}$, *j_l_*(*Qr_i_*) are the spherical Bessel functions and *Y_l,m_* the spherical harmonics. The translational diffusion *D_t_* is corrected for hydrodynamic and interparticle interactions in the same way as for the ZIF model according to Equation (4). The amplitude $A\left(q\right)$ describes the contribution to the intermediate scattering function from NM displacements with common relaxation time $\tau_{NM}$ of all modes *α* [22,50,51]
(7)$$A\left(q\right)=\sum_{\alpha }a_{\alpha }F_{\alpha }\left(q\right)/\left[F\left(q\right)+\sum_{\alpha }a_{\alpha }F_{\alpha }\left(q\right)\right]$$
and
(8)$$F_{\alpha }\left(q\right)=\langle \sum_{k,l}b_{k}b_{l}e^{iq\left(r_{k}-r_{l}\right)}\left(q∙e_{k}^{\alpha }\right)\left(q∙e_{l}^{\alpha }\right)\rangle ,$$
with mode eigenvector $e_{k}^{\alpha }$, formfactor $F\left(q\right)$, mode amplitude $a_{\alpha }$ and 〈∙〉 representing the orientational average.

Based on the EOM structures translational *D*_0,t_ and rotational *D*_r_ diffusion constants were determined by HYDROPRO [63]. The EOM structural ensembles were used to calculate respective diffusion coefficients and corresponding $S_{l}\left(Q\right)$ allowing to obtain the population weighted rotational diffusion coefficient averages (see Appendix A). For each EOM structure the first 10 nontrivial modes within an anisotropic network model (ANM) [64] were calculated and weighted equally for $F_{\alpha }\left(q\right)$ determination. $A\left(q\right)$ for the EOM ensembles were averaged according to respective populations. Finally, NSE data were fitted using the *D*_0,t_-values from 0.5% *w*/*v* DLS measurements as fixed input parameter and a common mode amplitude ~$a_{\alpha }$ reported as root mean square displacement (RMSD) of the mode displacements, the hydrodynamic function *H* and a common NM relaxation time $\tau_{NM}$ a free fit parameters (see Figure 5 and Figure 6C,D for the respective values). We obtain an NM relaxation time of $\tau_{\mathrm{NM}}\approx 7\;\mathrm{ns}$ that is approximately independent of GndCl concentration, and RMSD values that increase with GndCl concentration demonstrating a higher protein flexibility that is correlated with structural expansion of MBP. Corresponding NM fits to the NSE data are shown in Figure 4C,D for 0.2 M and 6 M GndCl and Appendix A for all GndCl concentrations. The quality of the NM fits is comparable, if not slightly better, than the fits using the ZIF model (see Appendix A) as evidenced by the $\chi^{2}=4.2$ value for the fit of the entire NSE data set and all GndCl concentrations.

*D*_0,t_ of the EOM ensemble are in good agreement with the 0.5% *w*/*v* DLS measurements showing the largest deviation for 0.2 M GndCl (theoretical: 5.33 Å2/ns versus DLS: 4 Å2/ns). We observe again $H_{short}>H_{long}$ except for 0 M and 6 M GndCl where both values are equal. The largest deviations in $H_{short}$ and $H_{long}$ for MBP in 0.2 M GndCl are related to the most compact structure of MBP that was already observed by the increased *R*_H_/*R*_G_-value for this sample (see Figure 2D). It should be noted that the relaxation process at long times is mainly determined by $D_{0,t}H$. Deviations in $D_{0,t}$ will be compensated directly by *H*. The rotational correlation times τ_r_ = 1/6*D*_r_ of the EOM ensembles are in general much smaller than the $\tau_{p=1}$ of the Zimm model (e.g., 0 M GndCl τ_r_ ≈ 47 ns, $\tau_{p=1}$ ≈ 73 ns; 6 M GndCl τ_r_ ≈ 84 ns, $\tau_{p=1}$ ≈ 183 ns). The rotational diffusion contributes here most to the long-time relaxation for *t* > 20 ns which allows a good separation from the internal mode spectrum at *t* < 20 ns, which is also visible in the NSE data (see Figure 4 and Appendix A). The NM relaxation time with $\tau_{\mathrm{NM}}\approx 7\;\mathrm{ns}$ is roughly equal for all GndCl concentration. The mode RMSD shows smaller values for GndCl concentrations of 0 and 0.2 M, which seems to be related to the more compact structure of MBP as discussed in the previous section. For less compact and more extended structures the local displacements are found to be larger.

To relate RMSD and τ to friction we may look at the model of a Brownian oscillator. For a Brownian oscillator the RMSD and relaxation time τ are related to the force constant *k*, friction ζ and thermal energy $k_{B}T$ by $RMSD^{2}=k_{B}T/k$ and τ=ζ/k. This leads to the following connection
(9)$$\zeta =k_{B}T∙\tau /RMSD^{2}$$

Thus, the increasing RMSDs as function of GndCl and the approximately constant τ as obtained from the NM fits are related to a decreased friction ζ with increasing GndCl concentration (see Figure 6B). This behavior is analogous to the observed decrease of $\tau_{i}$ using the ZIF model (compare Figure 6A). Both ZIF and NM data analysis approaches reveal the loss of friction within the MBP chain due to structural expansion. The differences in the absolute values and units of the NM-based approach and the ZIF model are related to the conceptual differences of the two mathematical methods, particularly that the first ZIF mode incorporates rotational diffusion contributions that are slower than the observation time of NSE, while for the NM-based approach a chain performing rigid-body rotational and translational diffusion is assumed that includes internal displacement patterns along the NM directions.

## 3. Conclusions

To conclude, we could show that MBP is in a partially folded structure in aqueous solution at neutral pD, which corroborates previous observations concerning MBP structure under acidic solvent conditions [22]. Furthermore, we demonstrate that 0.2 M GndCl results in charge screening effects in MBP which have strong implications on MBP structure. Charge screening mitigates the electrostatic repulsive constraints resulting in attractive intra-chain interactions, which leads to structural collapse of MBP. In contrast, at higher GndCl concentrations with *c*_GndCl_ ≥ 1 M, the presence of GndCl reverses the stabilizing effects and leads to a fully expanded protein. Repulsive intra-chain interactions between the protein chain emerge and cause MBP to attain a swollen conformation. This is supported by the structure factors, which show a crossover from attractive inter-particle interactions below 1 M GndCl to repulsive ones above. Furthermore, CD experiments demonstrate that MBP remains partially folded irrespectively of the GndCl concentration. GndCl concentrations above 2 M GndCl result in the emergence of the spectroscopic signature of PPII helix that is formed at the expense of α-helical or β-sheet structures. Structural expansion and the onset of the CD signal for PPII helix formation in MBP have been found to be correlated.

Dynamics of MBP have been investigated using NSE. We could demonstrate in our work that structural perturbations of MBP by GndCl result in significant changes of internal protein dynamics. In general, we observed deviations of MBP dynamics from ideal-polymer-like behavior for all GndCl concentrations. For MBP in the structurally most compact states at 0 M and 0.2 M GndCl we found that motions in MBP are strongly dominated by internal friction. On the other hand, a transition to more solvent friction driven dynamics is observed for the denatured states of MBP approaching a limiting internal friction with a relaxation time of around $\tau_{i}=40\;\mathrm{ns}$. In contrast to this observation, motions of highly denatured myoglobin (Mb) in 3 M GndCl have been observed to follow ideal Zimm dynamics with $\tau_{i}=0\;\mathrm{ns}$ [65]. While Mb has a similar molecular mass as MBP, it contains a significantly larger fraction of hydrophobic residues than the intrinsically disordered MBP. While GndCl acts as a good solvent for Mb resulting in ideal Zimm-like dynamics, the larger number of hydrophilic and charged amino acids in MBP prevent the protein from following ideal polymer-like dynamics at high denaturant concentrations. Structural expansion and semi-flexibility as seen by the CD signature for PPII helix most likely also lead to internal friction in MBP at high GndCl concentrations.

Using a NM based approach as alternative for NSE data interpretation we could demonstrate that relaxation times of internal modes remain effectively invariant of GndCl concentration changes, while an increase of protein flexibility in terms of RMSD values is correlated with structural expansion and eventually the formation of PPII helix. Importantly, both the polymer-based ZIF and the structure-based NM NSE data analysis approaches led to the identical conclusion: Structural expansion of MBP results in the loss of friction in the protein chain. However, the conceptual difference between the NM approach and the ZIF model is twofold: Primarily, the ZIF model assumes a flexible chain that reconfigures itself to reproduce the Gaussian-like configuration, while the NM method assumes an ensemble of distinct chain configurations performing rigid-body rotational and translational diffusion with small and fast internal NM displacements acting on short time and length scales. Second, related to the former aspect, in the ZIF model displacements of the different Zimm modes are fixed to $\sim R_{e}^{2}$ with a characteristic timescale τ_p_ = 1 + τ_I_, while for the NM approach internal displacement RMSD and relaxation time τ_NM_ are two independent parameters and are chosen complementarily to translational/rotational diffusion.

Taken together, all the aspects reported in this manuscript highlight the importance of structural and dynamical plasticity of IDPs in solution and their fundamental difference to the behavior of ideal polymers. Since under in vivo conditions MBP interacts with negatively charged myelin membranes that results in protein folding [20] and formation of concentrated liquid-like protein phases [24,25], significant changes in its dynamic behaviour under in vivo conditions are to be expected as evidenced by our in vitro studies. Perturbation of MBP dynamics by interaction with biological myelin membranes or by crowding conditions are exciting scientific topics that would merit further studies in the future allowing to gain more detailed insights on the relationship of in vitro and in vivo conditions.

## 4. Materials & Methods

### 4.1. Sample Preparation

Bovine MBP and all used chemicals were bought from Sigma Aldrich (St. Louis, MO, USA). MBP was used without further purification. For sample preparation MBP powder was dissolved in D_2_O (99.9% D atom) buffer (50 mM Na_2_HPO_4_/NaH_2_PO_4_, pD 7.0) in different concentrations of D_2_O-exchanged guanidinium chloride (GndCl). For samples used in SANS and NSE experiments, D_2_O-exchanged MBP was used instead. For that purpose, MBP was dissolved in D_2_O, incubated for a few hours, and rapidly frozen using liquid nitrogen. The frozen MBP solution was then freeze-dried overnight. The GndCl concentrations used were 0 M, 0.2 M, 1 M, 4 M, and 6 M. All buffer solutions were adjusted to pD 7.0 by adding DCl/NaOD. Protein concentration was determined using UV/Vis absorption at 280 nm with a nanodrop instrument (Nano-Drop 2000c, Thermo Scientific, Waltham, MA, USA) using an extinction coefficient of *E*_1%_ = 5.89.

The D_2_O exchange procedure of GndCl was started by weighing 300 g of the GndCl powder (99% purity) and dissolving it in 400 mL D_2_O. It was then incubated for approximately 15 h. The next step was to evaporate the D_2_O by using a rotary evaporator. Rotation speed of the rotary apparatus was 105 rpm, and temperature of the thermal bath 323 K. Vacuum was initially set to 800 mbar and then lowered to 40 mbar within 5 h. Once the D_2_O was mostly evaporated and the solution became visibly cloudy, the rotation was stopped. The partially deuterated GndCl was re-dissolved in 400 mL D_2_O and left for another 15 h. Afterwards, the procedure was repeated twice. In the last repetition, when the wet powder became visible, the pressure was reduced to 15 mbar for an hour to ensure complete drying.

### 4.2. Experimental Methods and Data Analysis

#### 4.2.1. CD Experiments

CD was measured on a Jasco J-1100 spectrometer (Jasco, Tokyo, Japan) in the far-UV regime (190–260 nm). CD measurements were performed with a dismountable cuvette of 0.1 mm pathlength and volume of 80 μL. Protein samples with MBP concentration of 2.5 mg/mL and corresponding GndCl buffer solutions were measured at a temperature of 295 K. The measured ellipticity θ in units of mdeg was converted into mean residue ellipticity according to ${\left[\theta \right]}_{MR}=M\;\theta /10\;d\;c$, where *M* is the mean residual weight, *d* is the cuvette pathlength in cm, and *c* is protein concentration in mg/mL.

#### 4.2.2. DLS and Viscosity Measurements

DLS measurements were performed using a Zetasizer Nano-ZS (Malvern Panalytical, Malvern, UK). Protein volume used was 70 μL and MBP concentrations of 0.5% and 5% *w*/*v* were measured. The hydrodynamic radii were obtained according to the Stokes–Einstein equation
(10)$$R_{h}=\frac{k_{B}T}{6\pi \eta D},$$
where $k_{B}$ is the Boltzmann constant, η solvent viscosity at *T* = 295 K and *D* the translational diffusion coefficient of the 0.5% *w*/*v* MBP solutions measured by DLS. Sample temperature was *T* = 295 K.

The viscosities of the 0.5% and 5 % *w*/*v* protein and buffer solutions were measured using a Lovis 2000M falling ball microviscometer (Anton Paar, Graz, Austria). For each viscosity measurement 100 μL sample volume was used.

#### 4.2.3. SAXS/SANS Experiments and Data Analysis

SAXS was measured on an in-house SAXS instrument Ganesha (Xenocs, Grenoble, France) located at JCNS-1/IBI-8. High flux configuration and two detector distances were chosen to cover the *q*-range between 0.01–0.7 ${Å}^{-1}$. Measured protein concentrations by SAXS were 0.5% and 5% *w*/*v*. SANS was measured on the small-angle neutron diffractometer KWS-2 located at MLZ, Garching, Germany [35,36,37]. Two detector distances of 4 m and 8 m were used in combination with neutron wavelengths of 4.66 Å and 7 Å. Protein concentrations measured with SANS were 5%, 1% and 0.5% *w*/*v*. The scattering vector *q* is defined in this work as *q* = 4π/λ*sin(θ/2) with the incident X-ray or neutron wavelength λ and the scattering angle θ.

Guinier analysis of SAXS/SANS form factors $P\left(q\right)$ has been performed according to $ln\left[P\left(q\right)\right]\sim -R_{Guinier}^{2}q^{2}/3$ in the limiting range of $q_{max}R_{Guinier}\leq 1.1$ for the expanded states of MBP and $q_{max}R_{G}\leq 1.3$ for the compact state of MBP in 0.2 M GndCl, where $q_{max}$ is the maximal *q*-vector that fulfils the limiting Guinier criterium [41]. Furthermore, SAXS/SANS form factors were fitted by generalized Gauss functions according to [58,59,66]
(11)$$P\left(q\right)=\frac{1}{\nu U^{\frac{1}{2\nu }}}\gamma \left(\frac{1}{2\nu },U\right)-\frac{1}{\nu U^{\frac{1}{\nu }}}\gamma \left(\frac{1}{\nu },U\right),$$
with
(12)$$U=\left(2\nu +1\right)\left(2\nu +2\right)q^{2}R_{g}^{2}/6$$
and
(13)$$\gamma \left(a,x\right)=\int_{0}^{x}t^{a-1}\exp\left(-t\right)\mathrm{dt}.$$

Equation (11) is a generalization of the classical Debye function and yields as free fit parameters the radius of gyration $R_{G}$ and the scaling exponent ν.

Additionally, the SAXS form factors $P\left(q\right)$ have been fitted using the worm-like chain model describing semiflexible polymers with excluded volume interactions as derived by Pedersen and Schurtenberger [44] and considering corrections given by Chen et al [67]. The model by Pedersen and Schurtenberger takes contour length *L* and scattering length density (SLD) of both protein and solvent as fixed parameters; free fitted parameters are the segment length (Kuhn length) *l*_K_, and the cylinder radius *R*.

The software suite SasView version 5.0.4 was used for data fitting [68]. Mathematical models and numerical algorithms as implemented in the SasView software package have been used for SAXS/SANS data analysis [68].

The EOM algorithm is part of ATSAS package developed by EMBL Hamburg [49]. The algorithm uses the primary protein sequence in FASTA format as input. It first generates a pool of 10,000 possible structures and a conformational ensemble is selected which is then cross validated against the SAXS form factor. The results are representative of the conformational ensemble of the disordered protein in solution.

The structure factor *S*(*q*) at a concentration of 5% *w*/*v* was obtained by dividing the measured SANS or SAXS intensity of a 5% *w*/*v* MBP solution by the corresponding intensity of a diluted 0.5% *w*/*v* solution. The experimentally determined structure factors were fitted with either a theoretical model function for non-charged hard-spheres with excluded volume interactions [69] or the Hayter–Penfold rescaled spherical approximation for charged spheres with excluded volume interactions [70,71]. Fit have been done using SasView using the implemented algorithms [68].

#### 4.2.4. NSE Spectroscopy Experiments

Dynamics of MBP in 0 M and 0.2 M GndCl were measured on the NSE spectrometer IN15 at the ILL, Grenoble, France [52]. MBP samples in 1 M, 4 M, and 6 M GndCl were measured on J-NSE Phoenix at MLZ, Garching, Germany [53], J-NSE data reduction has been done with the DrSpine software [72]. On both instruments, three incident neutron wavelengths of 6, 8, and 10 Å were used during the experiment. The protein concentration measured was 5% *w*/*v* for all GndCl concentrations, and sample stability was monitored with in-situ DLS on both instruments [73]. Fits to the NSE data with stretched exponential fits, ZIMM model, ZIF model and NM analysis were done using the Python package Jscatter 1.4.0 [74].

## Figures and Tables

**Figure 1 ijms-23-06969-f001:**
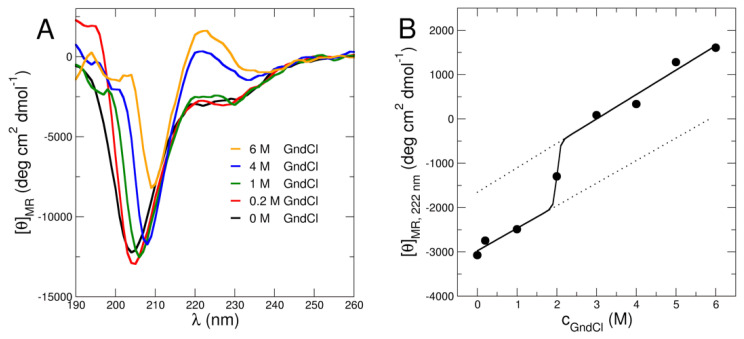
(**A**) CD signal at different GndCl concentrations demonstrating the largely disordered nature of MBP under all investigated solvent conditions. The emerging peak between around 220 and 230 nm is indicative of polyproline type II helix formation with accompanied structural expansion of MBP. (**B**) CD intensity at a wavelength of 222 nm as function of c_GndCl_ reporting on structural expansion of MBP. Symbols show experimental CD data, the solid line indicates an empirical sigmoidal fit and the dotted lines show pre- and post-transitional sloping behavior.

**Figure 2 ijms-23-06969-f002:**
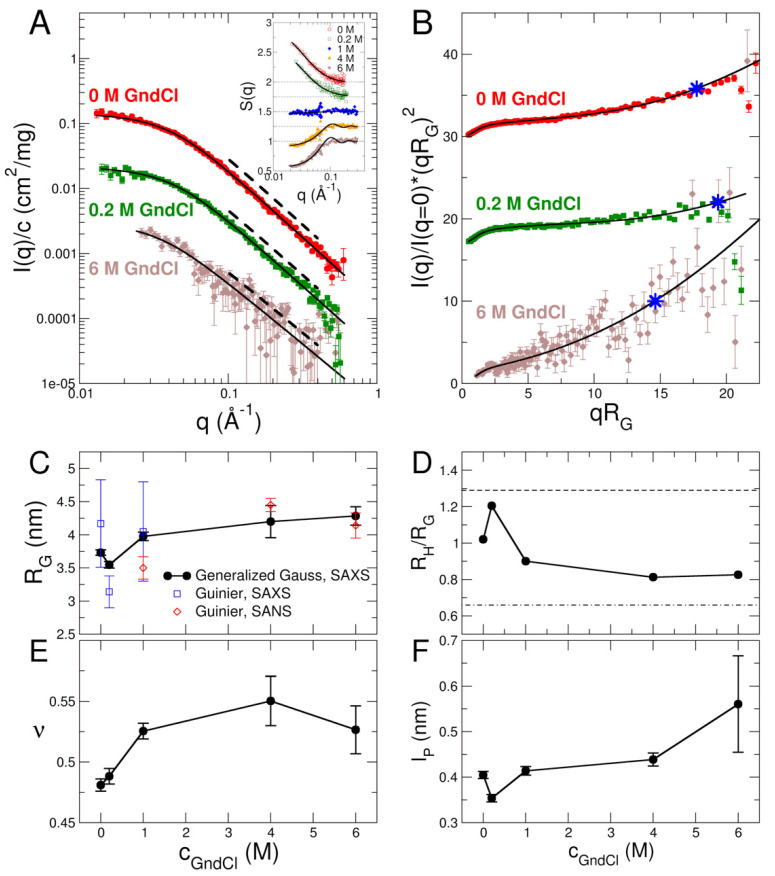
Structural information obtained with SAXS. (**A**) Form factors of native MBP in 0 M GndCl, the structurally collapsed state in 0.2 M GndCl, and the fully denatured and expanded state in 6 M GndCl. Symbols represent experimental SAXS data, solid black lines are fits with generalized Gauss functions, dashed lines indicate power law scaling behavior of the data. The inset in panel A shows the structure factors of the GndCl concentration series (empty symbols: SAXS, filled symbols: SANS data). Structure factors are shifted by an offset of 0.25 for clarity. (**B**) Kratky plots. Symbols indicate experimental SAXS data, solid black lines represent fits with a worm-like chain model. All data start from the origin but are shifted for clarity. Blue asterisk * corresponds to the length scale of the stiff segment given by *q** = 1.91/*l*_p_. (**C**,**E**) Radius of gyration *R*_G_ and scaling exponent ν as function of GndCl concentration. Black filled dots are obtained from generalized Gauss fits of SAXS data; empty symbols are *R*_Guinier_ values obtained from Guinier fits of SAXS and SANS data. (**D**) Influence of GndCl on the compactness of MBP in terms of *R*_H_/*R*_G_ ratio. Dashed lines indicate extreme limits for a globular and random coil-like protein. (**F**) Modulation of persistence length *l*_p_ of MBP by GndCl.

**Figure 3 ijms-23-06969-f003:**
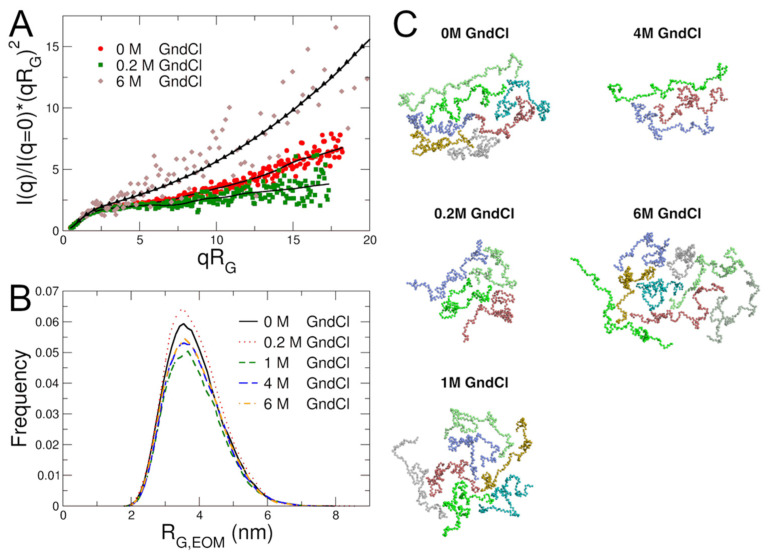
Results from EOM analysis. (**A**) Experimental SAXS data at different GndCl concentrations and fits using EOM modelling. Symbols represent experimental data; solid black lines are theoretical fits using EOM modelling; black triangles indicate the fitted generalized Gauss curve that was used as input for the EOM simulation of MBP in 6 M GndCl. (**B**) Radii of gyration distribution of MBP obtained from EOM analysis as function of GndCl concentration. (**C**) Coarse-grained structural models of MBP at different GndCl concentrations that appeared most frequently during EOM modelling. Structural models were used as representations of the conformational ensemble of MBP and as test models for normal mode analysis of NSE data.

**Figure 4 ijms-23-06969-f004:**
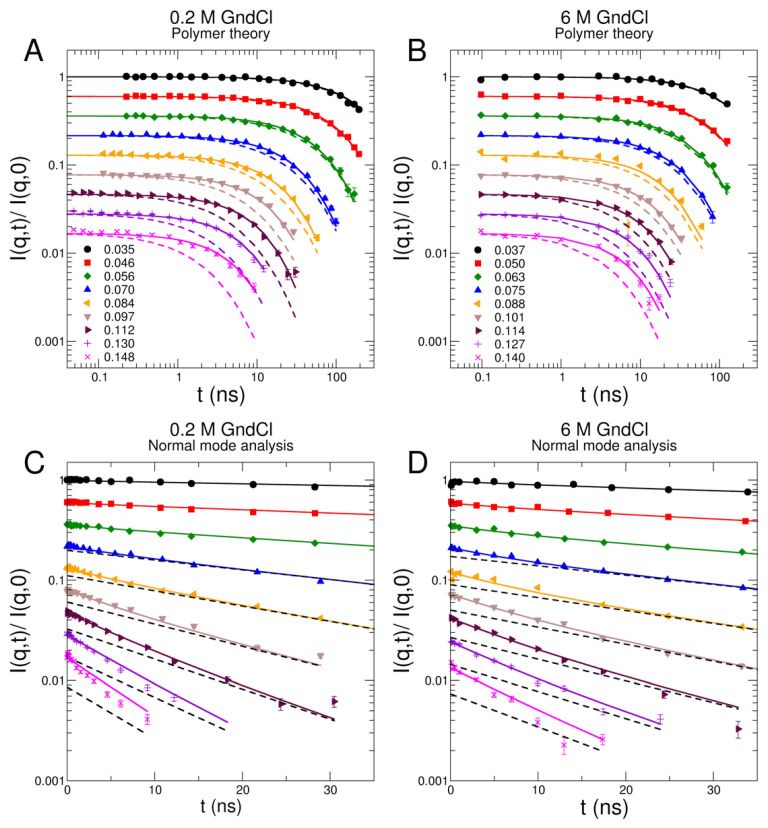
NSE data of MBP in representative GndCl concentrations. (**A**,**C**) MBP in D_2_O buffer solution with 0.2 M GndCl, (**B**,**D**) structurally expanded MBP in 6 M GndCl. (**A**,**B**) NSE spectra fitted with Zimm (dashed lines) and ZIF (solid lines) polymer models. Experimental data are represented by symbols. The scattering-vectors of the NSE spectra are given in units of Å^−1^ in the legends. (**C**,**D**) NSE spectra interpreted using normal mode analysis (solid lines) of the course-grained structural ensemble shown in Figure 2C. The black dashed lines indicate global translational and rotational diffusion of the structural models. All spectra start at unity. They are shifted by a factor of 0.8 for clarity.

**Figure 5 ijms-23-06969-f005:**
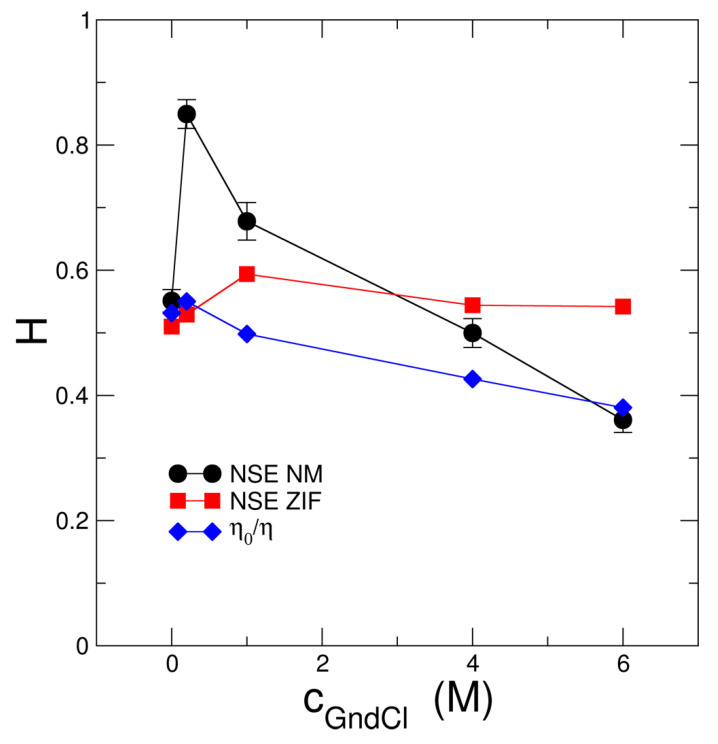
Hydrodynamic function $H_{short}$ obtained from fits of the NSE data using the ZIF model (NSE ZIF) and NM analysis (NSE NM) as compared to macroscopic hydrodynamic behavior as determined by the relation $H_{long}=\eta_{0}/\eta$ of $\eta_{0}$ buffer and η protein solutions.

**Figure 6 ijms-23-06969-f006:**
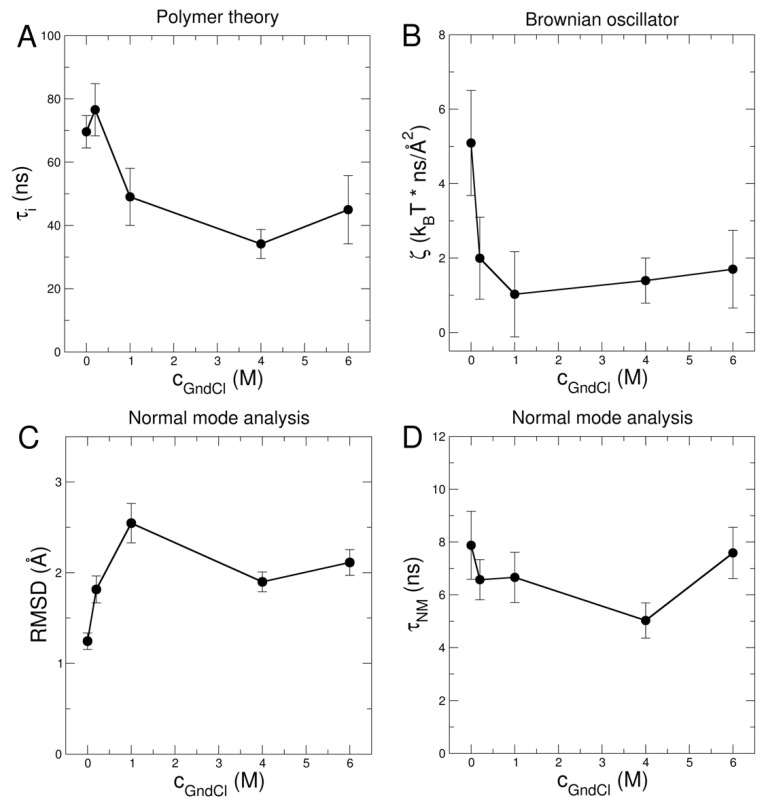
(**A**) Relaxation time of internal friction $\tau_{i}$ obtained from the ZIF polymer model and (**B**) friction ζ as determined within the model of the Brownian oscillator. (**C**) RMSD per atom and (**D**) relaxation times of internal modes as function of GndCl concentration as determined by using NM analysis of the NSE data.

## Data Availability

Experimental IN15 NSE data is available under doi:10.5291/ILL-DATA.CRG-2803; J-NSE, SANS and SAXS data under doi.org/10.5281/zenodo.6549012.

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
