# Peer review of "Variation of Structural and Dynamical Flexibility of Myelin Basic Protein in Response to Guanidinium Chloride"

_ijms, 2022, doi:10.3390/ijms23136969_

Round 1

Reviewer 1 Report

The manuscript "Variation of Structural and Dynamical Flexibility of Myelin Basic Protein in Response to Guanidinium Chloride " is a well designed study that presented a wide range of complementary experimental techniques. This study shows good agreement between theory and experiment and clearly demonstrates results, that are discussed extensively and clearly. It is recomended for publication, there are however little remrks:

- The importance of conducting all experiments in D2O should be explained;

-  Paragraph 2.2, lines 264-266: this is the only time the authors describe the instruments used within the Discussion section. All other instruments are (correctly) described in the Materials and Methods section.

- In the bibliography section references are not always presented homogeneously: year is most times bold, sometimes not; journal names are some times extended, sometimes abbreviated. Please uniform the style.

Author Response

First of all, we would like to thank the editor for taking care of our manuscript and both referees for the appreciation of our work. We would like to thank you for your time that you have invested to review the manuscript and for your helpful and positive comments that allowed us to increase the quality of the manuscript. In the following we describe the changes made in the manuscript. Changes following to comments by referees 1 and 2 are marked in the word document using the “track changes mode”.

Reviewer 1

The manuscript "Variation of Structural and Dynamical Flexibility of Myelin Basic Protein in Response to Guanidinium Chloride " is a well designed study that presented a wide range of complementary experimental techniques. This study shows good agreement between theory and experiment and clearly demonstrates results, that are discussed extensively and clearly. It is recomended for publication, there are however little remrks:

Comment 1:

- The importance of conducting all experiments in D2O should be explained;

Reply to comment 1:

Thank you for the appreciation of our work and your comments. It is certainly not clear to all readers why the experiments have been performed in D2O solvent. The reason is that neutron scattering experiments require the usage of deuterated solvents for technical reasons. Performing e.g. neutron spin-echo (NSE) experiments in H2O-based buffers is not feasible and deuterated solvents must be used. For stringent consistency throughout our study all experiments have been performed in deuterated solvent.

Changes to the manuscript:

We added the following sentence to the manuscript on page 2:

For neutron scattering experiments - and in particular for NSE studies - deuterated solvents are required to reduce the incoherent scattering contribution of the background and to maintain the polarization of the neutron beam in the NSE instrument. For consistency all experiments have been performed at a sample temperature of 295 K and in D2O buffer (99.9% D atom, 50 mM Na2HPO4/NaH2PO4, pD 7.0) containing additionally different concentrations of the denaturant GndCl.

Comment 2:

-  Paragraph 2.2, lines 264-266: this is the only time the authors describe the instruments used within the Discussion section. All other instruments are (correctly) described in the Materials and Methods section.

Reply to comment 2:

Yes, indeed we forgot to include the names and references of the SAXS and SANS instruments in the Results and Discussion section. Thank you for carefully reading our manuscript and to find that missing statement.

We have decided not so to provide the names and manufacturers of the used CD, DLS and microviscosimetry instruments in the Results and Discussion section as the detailed information can be found in the Materials & Methods section.

Changes to the manuscript:

The following sentence has been added on page 3 of the manuscript.

X-ray scattering experiments have been performed using a lab-based Ganesha SAXS instrument (Xenocs, Grenoble, France) located at JCNS-1/ IBI-8, FZJ ,Jülich, Germany and SANS was measured on the small-angle neutron diffractometer KWS-2 located at MLZ, Garching, Germany [35–37].

Comment 3:

- In the bibliography section references are not always presented homogeneously: year is most times bold, sometimes not; journal names are some times extended, sometimes abbreviated. Please uniform the style.

Reply to comment 3:

Yes, thank you very much. This is indeed true and has escaped our attention. References have been uniformed in style.

Changes to the manuscript:

References have been updated (all publication years in bold and journal names are spelled out).

Reviewer 2 Report

The article presents the structural variations of a model intrinsically disordered protein using different methodologies, also in the presence of guanidium chloride.

The study is well planned and accurately presenters. The authors fails however to present the relationship between the in vitro conditions used and the in vivo ones.

I suggest testing in the presence of glycerol.

Author Response

First of all, we would like to thank the editor for taking care of our manuscript and both referees for the appreciation of our work. We would like to thank you for your time that you have invested to review the manuscript and for your helpful and positive comments that allowed us to increase the quality of the manuscript. In the following we describe the changes made in the manuscript. Changes following to comments by referees 1 and 2 are marked in the word document using the “track changes mode”.

Reviewer 2

The article presents the structural variations of a model intrinsically disordered protein using different methodologies, also in the presence of guanidium chloride.

Comment 1:

The study is well planned and accurately presenters. The authors fails however to present the relationship between the in vitro conditions used and the in vivo ones.

Reply to comment 1:

Thank you very much for the appreciation of our work. We added some explaining sentences to the Conclusion to connect our in vitro observations with in vivo conditions.

We would like to mention, however, that NSE experiments that we used in our study to probe protein dynamics are unfortunately not suitable for in vivo studies of entire oligodendrocytes where MBP is found in the cytoplasm. Scattering techniques always require isolated pure protein solutions as otherwise the ensemble average of an entire cell is probed by the experiment and unfortunately no information can be deduced on the individual behavior of MBP.

Our study was intended to investigate the structural and dynamical response of MBP to changes of GndCl concentration (in vitro) and it is in this sense a contribution to the characterization of the physico-chemical behaviour of MBP as a representative of an intrinsically disordered protein.

Changes to the manuscript:

We added the following sentences to the Conclusion on page 15.

Since under in vivo conditions MBP interacts with negatively charged myelin membranes that results in protein folding [20] and formation of concentrated liquid-like protein phases [24,25], significant changes in its dynamic behavior under in vivo conditions are to be expected as evidenced by our in vitro studies. Perturbation of MBP dynamics by interaction with biological myelin membranes or by crowding conditions are exciting scientific topics that would merit further studies in the future allowing to gain more detailed insights on the relationship of in vitro and in vivo conditions.

Comment 2:

I suggest testing in the presence of glycerol.

Reply to comment 2:

Thank you for that suggestion. First experiments on the effect of glycerol on the dynamics of MBP have been performed in 2021 using the wide-angle spin-echo spectrometer WASP located at the research reactor of the Institut Laue-Langevin (ILL) in Grenoble, France. Experimental WASP data are currently being analyzed. The experimental WASP data set is, however, not complete yet and further experiments are needed to gain a full picture on the changes in internal dynamics due to variation of buffer viscosity. The research reactor of the ILL is unfortunately not operational in 2022 and will restart only in 2023. Therefore, we cannot present any results on the effect of glycerol on the dynamics of MBP in the present manuscript.

Our aim in the present study was to investigate the effect of the denaturant GndCl on the structure and dynamics of MBP and we believe that addition of further experimental results of the effect of glycerol on MBP is in fact not needed for the present study.

Changes to the manuscript:

None